# Protein-protein interactions enhance the thermal resilience of SpyRing-cyclized enzymes: A molecular dynamic simulation study

Qi Gao, Dangling Ming  *

College of Biotechnology and Pharmaceutical Engineering, Nanjing Tech University, Nanjing City, Jiangsu, PR China

* dming@njtech.edu.cn

## Abstract

Recently a technique based on the interaction between adhesion proteins extracted from *Streptococcus pyogenes*, known as SpyRing, has been widely used to improve the thermal resilience of enzymes, the assembly of biostructures, cancer cell recognition and other fields. It was believed that the covalent cyclization of protein skeleton caused by SpyRing reduces the conformational entropy of biological structure and improves its rigidity, thus improving the thermal resilience of the target enzyme. However, the effects of SpyTag/ Spy-Catcher interaction with this enzyme are poorly understood, and their regulation of enzyme properties remains unclear. Here, for simplicity, we took the single domain enzyme liche-nase from *Bacillus subtilis* 168 as an example, studied the interface interactions in the SpyR-ing by molecular dynamics simulations, and examined the effects of the changes of electrostatic interaction and van der Waals interaction on the thermal resilience of target enzyme. The simulations showed that the interface between SpyTag/SpyCatcher and the target enzyme is different from that found by geometric matching method and highlighted key mutations at the interface that might have effect on the thermal resilience of the enzyme. Our calculations highlighted interfacial interactions between enzyme and SpyTag/Spy-Catcher, which might be useful in rational designs of the SpyRing.

## Introduction

Enzymes have been widely used in industry because of their superlative catalytic efficiency, stereoselectivity, mild reaction conditions and environmental friendliness, and more than a hundred enzymes have found large-scale industrial uses [1]. However, many promising enzymes developed in laboratory have encountered great challenges in industrial application, due to their poor stability at high processing temperatures [2], easy deactivation and difficult storage [3]. To meet these challenges, conventional methods, such as directed evolution, rational design and other structural or library based methods have been developed and gained

**Funding:** This work was supported, in part, by the National Key Research and Development Program

of China (Grant No. 2019YFA0905700, 2017YFC1600900) to DM. We are grateful to the High Performance Computing Center of Nanjing Tech University for supporting the computational resources. The funders had no role in study design, data collection and analysis, decision to publish, or preparation of the manuscript.

**Competing interests:** The authors have declared that no competing interests exist.

remarkable success [4–6]. Since enzymes vary greatly in structure and function, the above-mentioned methods of enzyme modification can only be carried out on a case-by-case basis, lacking a unified guidance scheme [7, 8].

In the past few decades, protein cyclization technology has developed into as a generic approach to improve the thermal stability of enzymes. In protein cyclization, the N- and C-termini and other unfolded regions of a protein are covalently tethered so as to increase the rigidity of the protein, leading to higher entropy barrier when unfolding [9–11]. Various protein cyclization strategies such as chemical synthesis [12], transpeptidation using sortase [13], intein-mediated splicing [14], have been reported to increase thermal resilience to a certain extent. Very recently, a novel cyclization method called SpyRing has been reported, which uses a strategy of spontaneous isopeptide bond formation mediated by SpyTag/ Spycatcher [15, 16]. In SpyRing, SpyTag and SpyCatcher, two polypeptides originally discovered in *Streptococcus pyogenes* [17], spontaneously interact with each other to form isopeptide bonds, leading to covalent cyclization of the target enzyme backbone. One of the advantages of this reaction is that it can be carried out under a variety of different conditions, including reduction or oxidation conditions and a fairly wide pH range [18, 19].

In the past few years, several research groups have reported the use of SpyRing technique to increase the thermal resilience of various enzymes. Schoene and colleagues showed that SpyRing increases thermal resilience of β-lactamase by more than 60°C, which is much higher than that achieved by conventional point mutation methods [20]. Wang and colleagues used SpyRing to cyclize the lichenase from *Bacillus subtilis* 168 and raised the optimum temperature by 5°C [21]. What is impressive is that the cyclized enzyme can still maintains 80% of its catalytic activity at 100°C, while the linear enzyme almost loses all its catalytic activity at this temperature. Wang and colleagues studied the thermostability and organic solvent tolerance of cyclized L-phenylserine aldolase, and reported that cyclization increased the half-life of the enzyme by 8.3 times at 70°C, and the $T_{50}$ increased by 10.3°C [22]. Very recently, Zhou and colleagues compared the performances of linear xylanase and cyclized xylanase, which has potential sustainable commercial development value in green energy manufacturing industry. They found that although SpyRing did not significantly increase the optimal temperature of the enzyme, it did enhance thermal stability, ionic stability and resilience to aggregation and freeze-thaw treatment without affecting its catalytic efficiency [23]. Si and colleagues used cyclization technique to enhance the thermal resilient performance of a firefly luciferase (FLuc) from *Photinus pyralis*, a reporter enzyme with many potential academical research and industrial applications [24]. The optimum temperature of the cyclized enzyme is 35°C, which is 10°C higher than that of the linear FLuc. In addition, they found the half-life $t_{1/2}$ (at 45°C) and the melting temperature $T_m$ of the cyclized enzyme increased by 2.4 times and 16.7°C, respectively. So far, SpyRing is mainly designed for monomer enzymes. It is considered that the introduction of SpyCatcher/SpyTag at protein termini should avoid the catalytic sites of the enzyme, so as to improve the thermal stability of the enzyme without affecting its catalytic ability. It was believed that introduction of SpyRing should not hinder chaperone-assisted protein folding/refolding or the cofactor assisted protein refolding [25]. The application of SpyRing in a multi-enzyme complex systems requires detailed consideration of the relative configuration of the SpyTag/SpyCatcher and the enzymes.

In addition, in order to broaden the application scope of SpyRing, different mutations have been made to SpyTag/SpyCatcher, so that the entire enzyme system gained some new functions. Cao and colleagues introduced more than 10 negative-charge mutations in SpyTag/SpyCatcher, with a net charge of -21, and obtained a sensitive system in which cyclization reaction can be adjusted by environment PH [26]. Liu and colleagues reported through rational design and directed evoluation that three mutations in SpyTag/SpyCatcher can lead to orthogonal

reactivity of proteins with high sequence similarity, and obtain other valuable properties such as high selectivity and inverse temperature dependence [27]. Keeble and colleagues obtained a series of mutants through a phage-display platform and rational design [28, 29]. Through the optimization of the docking and cyclization reaction, they obtained mutants that greatly speed up the reaction by an order of magnitude, and expanded the application of the system in detecting the interaction between enzymes and other biological structures.

Very recently, Tian and colleagues introduced a special peptide (LysGlyLysGlyLysGly) into the C-terminus of SpyCatcher and three other Lys/Arg mutations around the binding domain. They found that the mutant SpyCatcher was more effectively attached to the glyoxyl-agarose support, making it highly binding to the SpyTag fused protein [30].

When applying SpyRing, in addition to requiring that the active site of target enzyme is not blocked by SpyTag/SpyCatcher, mutations are only selected to optimize the cyclization reaction between SpyTag and SpyCatcher; their potential effects on target enzymes through interface interactions are often ignored. In a recent study of SnoopRing cyclized luciferase, we found that cyclization also introduced important interactions at the interface between the enzyme and Catcher/Tag peptides [31]. Molecular dynamic simulation combined with SDS-PAGE analysis showed that the docking process of SnoopRing with luciferase is regulated by the interaction between Lys22 in Tag and Lys557 in luciferase. The final compact structure from the docking involves electrostatic interaction between Arg17 in Catcher and Asp520 in luciferase, and other hydrogen bonds and van der Waals interactions. Based on these observations, we suspected that SpyRing-mediated cyclization should also introduce important interactions at the interface between target enzyme and SpyTag/SpyCatcher. In order to study the role played by the interface interactions, for simplicity, this study took a single-domain enzyme lichenase [21] as an example to study how the interface and its mutations affect the performance of the target enzyme, especially the thermal stability. We used molecular dynamics simulations to determine the protein-protein interface (PPI) between the enzyme and SpyTag/SpyCatcher, which to some extent can be regarded as a restricted protein-protein docking problem. Then we used PPI to investigate the regulation effect of a variety of SpyRing mutations on enzyme properties. Our calculations suggested a pathway involving the docking of SpyTag/SpyCatcher with the enzyme, highlighting the key residues that regulate interface interactions between enzyme and SpyTag/SpyCatcher, and their effect on enzyme properties such as thermal resilience.

## Methods

### Molecular dynamics simulation of the cyclized lichanse structure

In order to investigate the interaction between SpyTag/SpyCatcher and enzymes and the effect of protein-protein interface mutations on enzyme thermal stability, we conducted a series of simulations on the cyclized system and systems with a few designed mutants. The X-ray structure of β-1,3–1,4-glucanase (lichenase) from *Bacillus subtilis* 168 was taken from the protein data bank (PDB) [32] with entry code 3O5S [33] and a resolution of 2.2A˚. The starting structure of the wild-type SpyTag/SpyCatcher was built using the homology modelling program MODELLER [34] with the X-ray structure (PDB entry code 4MLI, 2.1A˚ resolution [19]) as a template, adding missing residues. As described in reference [21], the termini of the enzyme were connected to SpyTag/ SpyCather via a 6-residue GSGGSG linker. In the initial configuration, SpyTag/ SpyCather was placed in a position where it did not have any contact with the enzyme, except that it was connected to the enzyme through a linker (Fig 1).

All simulations were carried out using GROMACS [35] program. The gromos53a6 force field [36] was used to generate the topology files of the original complex and the mutants. The

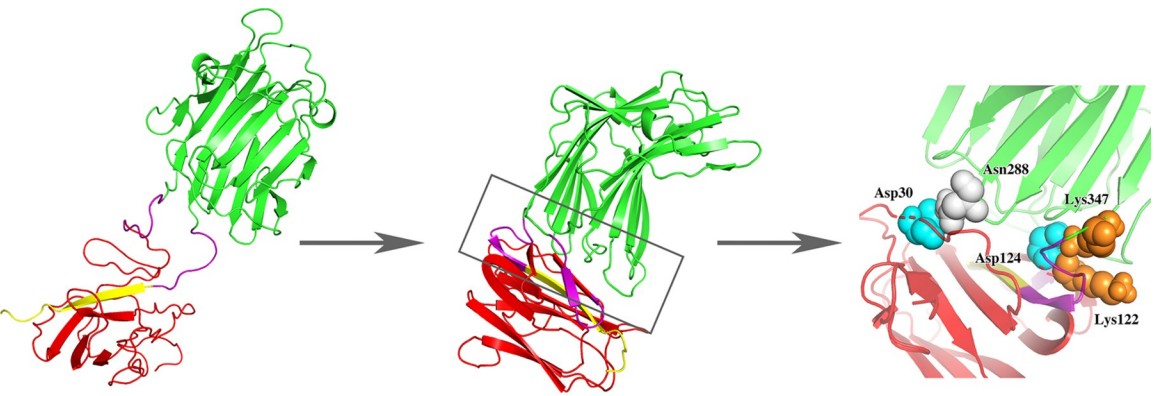

**Fig 1.** Protein structure of wild type before (left) and after (middle) simulated, where green is lichenase, yellow is SpyTag, and red and magenta are SpyCatcher (magenta highlights the part of SpyCatcher that interacts with the enzyme). The protein interface diagram (right) shows conservative interactions. The following figures apply the same colors to identify the enzyme and SpyCatcher/SpyTag.

SPCE water model was chosen so that the initial structure of each system was solvated in a cube of length 112.12 Å. An appropriate amount of sodium counterion was added to neutralize each system to keep it electroneutral. The system was first subjected to enery minimization using steepest descent method to eliminate large steric hindrance. Subsequently, with harmonic position restraints, two consecutive short equilibrium simulations with constant volume and constant pressure respectively were carried out to gradually bring the system to the target temperature of 333 K and pressure of 1atm. Finally, a product equilibration simulation of 100 ns was obtained. The van der Waals interactions were truncated at a cutoff distance of 1.0nm. Long range electrostatic interactions were handled using the particle mesh Ewald method (PME) [37] in reciprocal space, and the cutoff distance of real-space electrostatic interactions was set to 1.0nm. The time step was set to 2 fs, and trajectories were collected at 10ps intervals. The trajectories were analyzed using the GROMACS [35] built-in programs. The visualization programs VMD [38] and PyMOL [39] were used to examine the molecular dynamics trajectories and analyze interactions between the enzyme and SpyTag/SpyCather structure.

## Introductions of mutations to the cyclized lichenase

A visualization of the simulation system obtained from the simulation showed that a stable interface was formed between wild-type type SpyTag/SpyCather and the enzyme, and different types of interactions occurred at the interface, including electrostatic and hydrophobic interactions. The simulated trajectories were then examined and visualized using VMD, and for each snapshot, all amino acids within 5 Å of the enzyme were identified on SpyTag/ SpyCather. Given that these amino acids played an important role in the formation of the protein-protein interface between SpyTag/SpyCather and the enzyme, mutations at the interface might strongly affect the nature of the enzyme, including changes in enzyme thermostability. Here, we designed four types of mutations involving electrostatic interactions at the interface, and studied the effects of these mutations on the enzyme thermal resilience.

## Interactions at protein-protein interfaces

The detailed chemo-physical properties of the contacts between the enzyme and SpyCather/ SpyTag were determined using the software of Ring [40] with the default parameters thtat came with the program. The program was applied to the structures of snapshots uniformly

selected from trajectories produced by molecular dynamics simulations. Since this simulation was designed to link the separation state between the enzyme and SpyTag/SpyCather with the tight-packing state between them, the trajectories thus explored the evolution of protein-protein interactions (PPIs) during the formation of the interface. In particular, the simulations highlighted transient PPIs that occurs only in some intermediate snapshots but might play a role in inducing the formation of final interface. Some interactions, once formed, would stay there until the final formation of a stable PPI, while others might undergo a series of intermediate steps and eventually remained on the interface. Considering that in the SpyRing, both the receptor and the ligand proteins are usually negatively charged, leading to certain electrostatic interactions at the interface, and those mutations that involve change in charge may have significant impact on the system. Here, for the sake of simplicity, only the amino acids located at the interface were selected to design mutations to study the perturbation of the interface and its effect on the thermal resilience of the enzyme.

### Calculation of binding energy

To quantitatively evaluate the binding interaction between SpyTag/SpyCather and the enzyme and examine the perturbation caused by different mutations, the free energy calculations of the SpyRing system was carried out. The free energy change $\Delta G_{binding}$ of binding between protein (the enzyme) and ligand (SpyTag/SpyCather) in solvent can be expressed as follows:

$$\Delta G_{binding} = G_{complex} - (G_{protein} + G_{ligand})$$

where $G_{complex}$ is the total free energy of the protein-ligand complex, and $G_{protein}$, $G_{ligand}$ stand for the respective free energies of the protein and the ligand in solvent. Here, the program g_mmpbsa [41, 42] was used to analyze the difference in binding energies between wild type and mutants. Currently, the g_mmpbsa program does not include the calculation of the entropy term, therefore it cannot give the absolute binding energies. However, the tool is suitable for comparing the relative binding energies of different ligands that bind to the same receptor. Here, we treated lichenase as a protein receptor and SpyCatcher/SpyTag as the ligand.

### Comparison with random docking

What restrictions do the linkers at the two termini of the enzyme have on its docking with SpyTag/SpyCatcher? In order to answer this question, we compared docking based on SpyRing cyclization with random docking, which were derived using the program ZDOCK [43] without linking SpyTag/SpyCatcher to the enzyme. ZDOCK gave 2000 random conformations, of which the one with the highest score was selected for further optimization using RosettaDock [44, 45]. Then, after 10 cycles of re-docking and optimizing of the side chains, the protein-protein interface was identified and analyzed, and the binding energy was evaluated by Rosetta InterfaceAnalyzer [46].

## Results

### There is a stable interface between the enzyme and SpyTag/SpyCatcher

Three repeated simulations were performed on the SpyRing complex, in which the wild-type lichenase was cyclized with SpyTag/SpyCatcher. The simulations revealed the cyclization introduced significant interactions between the enzyme and the tag/catcher complex (Fig 1). The trajectories generated from MD simulations clearly showed that in the initial stage, the enzyme and the tag/catcher were separated, but after 20 ns, the two were tightly combined.

The simulation results revealed that the centroid distance between the enzyme and the tag/catcher was decreased from 4.8 nm to 3.2 nm, resulting in protein interface interactions, including electrostatic interactions, hydrogen bonding and hydrophobic interactions. Among the amino acids located at the interfaces, a subset was found to be conserved, including Asp30 near the N-terminus of the catcher, and Lys122, Asp124, located at the C-terminus of the catcher. The following mutation designs were then based on the protein-protein interactions observed at this interface.

## Mutations altered the intermediate pathways of docking between *SpyTag/SpyCatcher* and the enzyme, resulting in a change in the interface

**The Mutation from neutral to positively charged amino acid (T136K) had no significant effect on thermal resilience of SpyRing system.** The mutation T136K at the N-terminus of lichenase showed two random results. In the initial stage of the simulation, Lys136 and Asp124 had strong electrostatic interaction, making them close to each other. Later, they were separated again due to the electrostatic repulsion of Lys122 to Lys136 (Fig 2A). However, if Lys136 was electrostatically repelled by Lys122 in the initial stage, Asp124 cannot approach Lys136, leaving enough space near Lys136. In this case, Asp49 and Lys136 have a strong electrostatic interaction, which made the lichenase twisted to a certain extent (Fig 2B). However, this local conformational change and interaction did not seem to have a significant effect on the overall thermal stability of the SpyRing system, and the root mean square deviation (RMSD) of the mutant was basically the same as that of wild-type. The α-carbon root mean square fluctuations (RMSF) showed that the mutations did not decrease or even increased the fluctuation of amino acid residues, indicating that the mutations did not have a beneficial effect on the stability of the enzyme (S1 Fig).

**The mutation from positively charged to negatively charged amino acid (K122E) had no significant effect on thermal resilience of SpyRing system.** The mutation K122E was selected in SpyCatcher to introduce addition attraction at the interface between SpyTag/SpyCather and the enzyme, especially an attempt to introduce local interaction near Lys347 of the

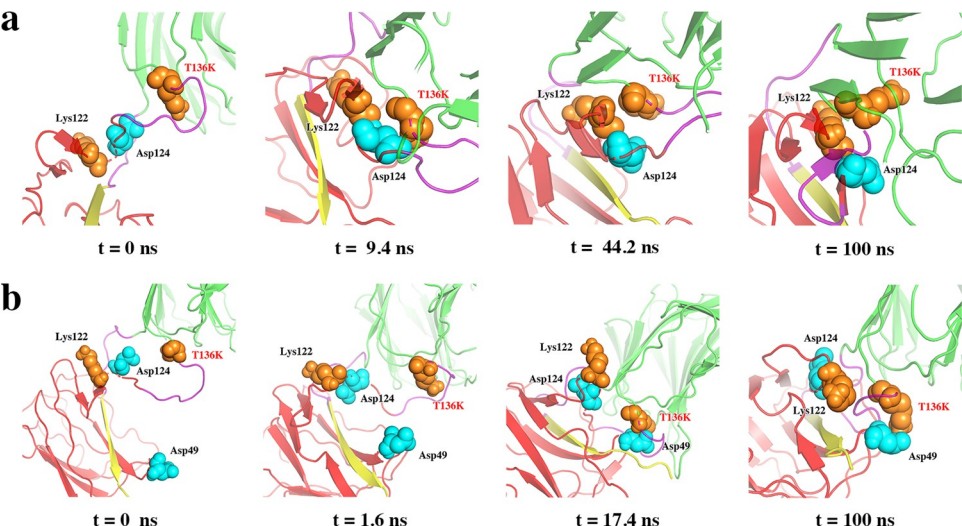

**Fig 2. Structure in simulated trajectory of T136K.** (a) and (b) are two simulation results. (a) Lys136 and Asp124 first approached each other and then separated again, which might be due to the electrostatic repulsion of Lys122. (b) It is not Asp124 but Asp49 and Lys136 form a strong electrostatic interaction and are close to each other.

enzyme. Indeed, in the initial conformation of the mutant K122E, both Glu122 and Glu34/Glu35 of SpyCatcher were found to be capable of forming electrostatic interactions with Lys347 of the enzyme. The simulation showed that at the beginning, both Glu122 and Asp124 quickly came into contact with Lys347 through electrostatic interactions. At this stage, Glu34 was also attracted to the vicinity of Lys347 because of its interaction with Lys347. However, at the end of the simulation, when the protein interface was finally formed, Glu34 was excluded from Lys347 due to the electrostatic repulsion of Glu122 (Fig 3A). A second simulation showed another possibility of electrostatic interaction with the key residue Lys347. In this case, the introduction of Glu122 by mutation K122E altered the local balance previously established between Lys347 and Asp124, and both Asp124 and Glu122 were pushed away from the vicinity of Lys37 because the third negatively charged Glu35 approached Lys347 and formed stronger electrostatic interaction. The space left behind was then occupied by Glu35, which maintained the resulting interface between SpyTag/SpyCather and the enzyme (Fig 3B). However, the oppositely charged mutation that tried to increase the local electrostatic interaction between SpyTag/SpyCather and the enzyme did not make the enzyme more stable, as shown by the fact that the RMSD of the mutant was basically the same as that of the wild type. The $\alpha$-carbon RMSF also showed that the mutation did not affect the fluctuation of most amino acid residues (S2 Fig).

**The mutation from positively charged to neutral (K122G) reduced RMSD of the enzyme.** In SpyRing, Lys348, a positively charged residue at the C-terminusof the enzyme, could form electrostatic attraction with Asp124 or Glu35 in SPyCather, and there was no obvious bias between the two according to the dynamic simulation. The mutation K122G near the C-terminus of SpyCather eliminated the positive charge at site 122 on SpyCather, eliminating the electrostatic repulsion with Lys348 and the electrostatic attraction of Glu35 at the same time, so that Lys348 can more tightly combine with Glu35 or Asp124(Fig 4). The mutation enhanced the enzyme's terminal amino acid Lys348 to participate in the electrical interaction at the interface, making the structure of the SpyRing complex more stable, and the dynamics of the main chain also showed a reduced RMSD. The $\alpha$-carbon RMSF suggested that the

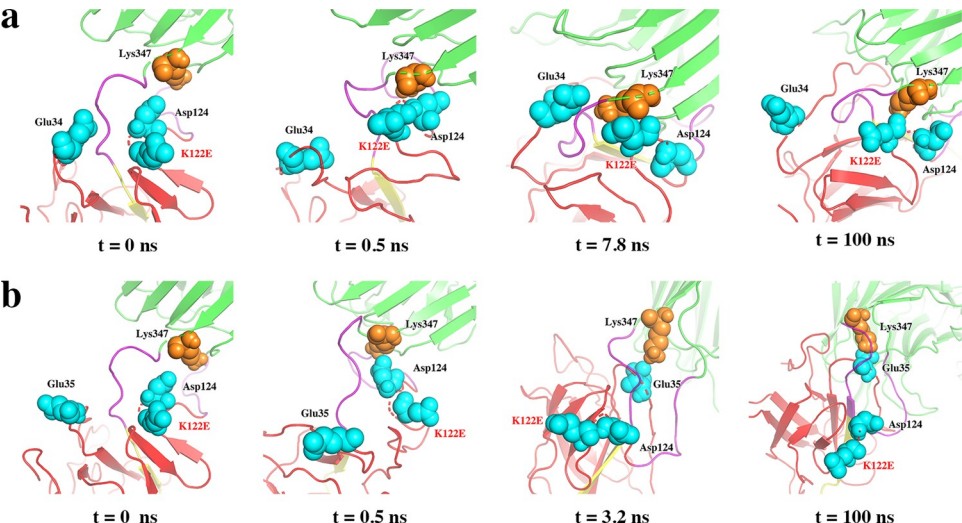

**Fig 3. Structure in simulated trajectory of K122E.** (a) and (b) are two simulation results. (a) Glu122 and Asp124 interacted with Lys347 first and lasted until the end; Glu34 was also very close to Lys347, but eventually left. (b) Both Glu122 and Asp124 were first closed to Lys347, and then left. On the contrary, Glu35 approached Lys347 later and remained in this state untill the end of the simulation.

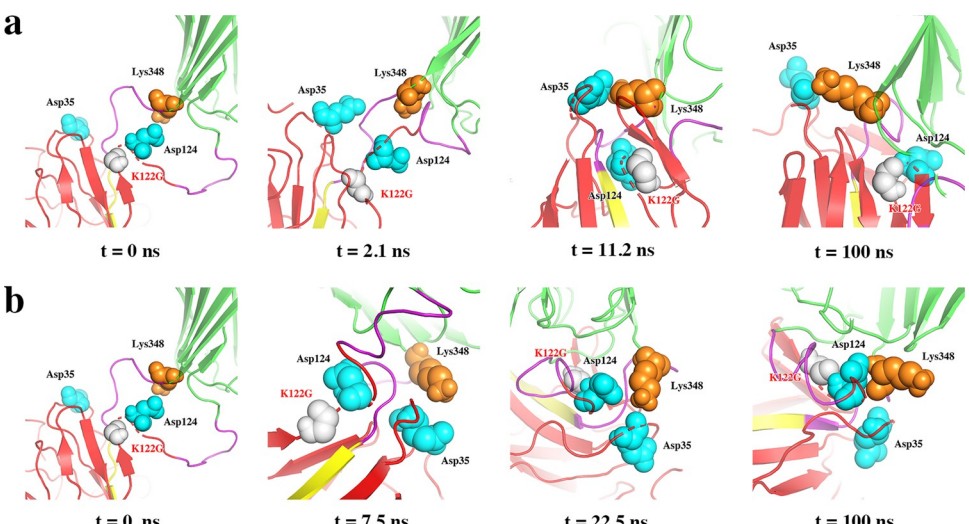

**Fig 4. Structure in simulated trajectory of K122G.** (a) and (b) are two simulation results. (a) Lys348 preferred Glu35 to Asp124 in terms of electrostatic attraction. (b) Lys348 preferred Asp124 to Glu35 in terms of electrostatic attraction.

mutation reduced the fluctuation of those amino acid residues particularly located at the active site, indicating that the mutation played a certain role in stabilizing the catalytic site (Fig 5).

**The mutation from negatively charged to neutral (D124G) reduced RMSD of the enzyme.** The D124G mutation at the C-terminus of SpyCather eliminated the negative charge of Asp124 originally present at the interface between SpyTag/SpyCather and the enzyme, thereby eliminating the electrostatic interaction between it and the positively charged Lys348 at the C-terminus of the enzyme. This made Lys348 form a local electrostatic interaction network with the two negatively charged amino acids Asp30 and Glu34 on SpyCather and the other positively charged amino acid Lys122on SpyCather. These four residues were in balance with each other, and conformations with different mutual orientations were observed (Fig 6). The final equilibrium dynamics simulation showed that this mutation resulted in the enhancement of local interaction at the interface, and the structure of the SpyRing complex was more stable, as revealed by the decrease in RMSD. On the other hand, the calculation of $\alpha$-carbon RMSF suggested that the mutation had no significant effect on the fluctuation of most amino acid residues (S3 Fig).

## Compared with random docking, the SpyRing method introduced a more stable interface

Compared with the wild type, the binding energy of all mutations had decreased, which meaned that the mutants are more stable than the wild type (before the mutation). The trend of electrostatic energy at the interface was basically the same as that of the calculated binding energy. In fact, electrostatic energy contributed the largest part of binding energy (Fig 7A). Among them, the most stable was K122G, which was consistent with the result that its RMSD had dropped significantly compared to other mutations. On the other hand, although the binding energies of two mutants T136K and K122E were lower than those of the wild type, their RMSD did not changed significantly. This might be because we only selected a subset of amino acids that interact at the interface to calculate the binding energy, while ignoring the large amount of solvent polar energy and electrostatic energy inside the protein.

However, when the enzyme was randomly docked with SpyTag/SpyCather without a linker as in SpyRing, we found that compared with the wild type, the binding free energy changes of

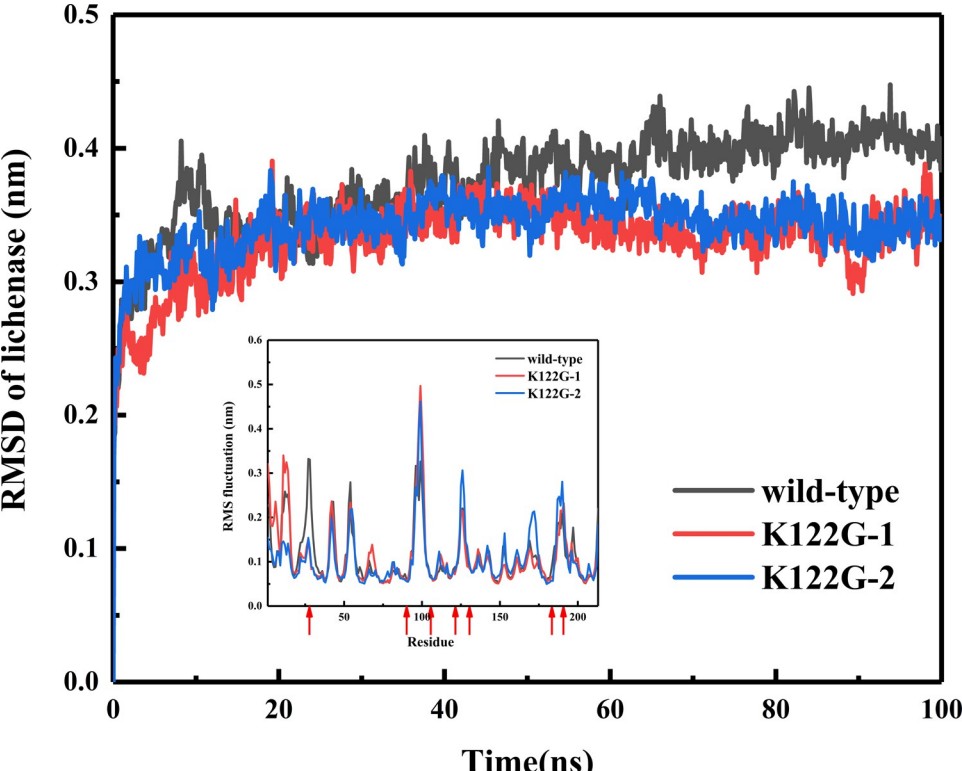

**Fig 5. RMSD and RMSF (the inset of Fig 5) of the lichenase with the mutation of K122G during 100 ns MD simulation.** The red arrows in the RMSF diagram refer to the active sites.

the mutants were more complicated, with some increasing and some decreasing (Fig 7B). This was in contrast to the SpyRing system, where the binding energy of all mutations was reduced. In SpyRing, the introduction of interface mutations had a great impact on the stability of the

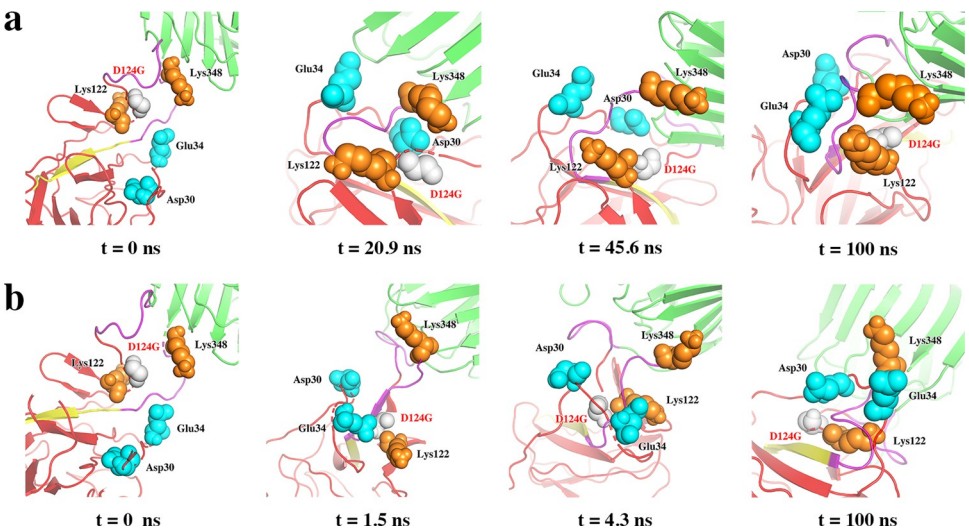

**Fig 6. Structure in simulated trajectory of D124G.** (a) and (b) are two similar simulation results. Lys348, Asp30, Glu34, Lys122 were connected to each other through electrostic interactions, and all four members maintained a balanced network structure.

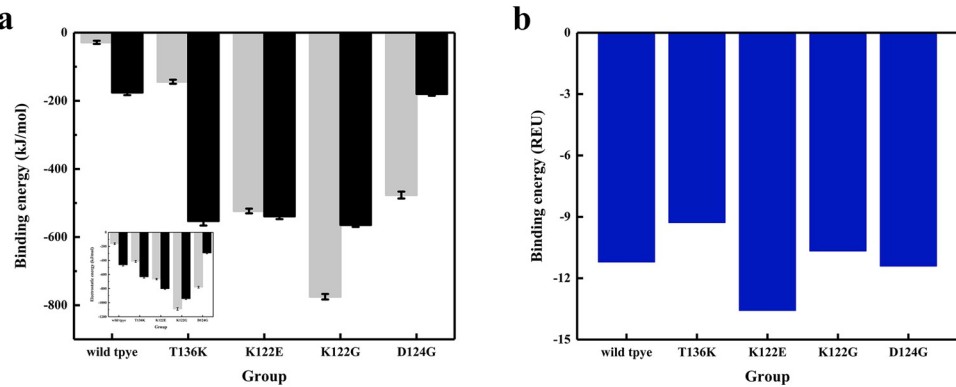

**Fig 7.** (a) Binding energy and electrostatic energy of wild-type and mutants, where the gray and the black bars are the results of two simulation calculations. (b) The binding energy of the complexes formed by random docking. REU is the Rosetta energy unit.

cyclase complex. However, the mutations selected in the interface of SpyRing complex were not necessarily on the interface of randomly docked complexes, which may explain the difference in binding energy between the two.

In terms of the composite structure, the relative orientation of the enzyme in the SpyTag/SpyCathcer and SpyRing complex is significantly different from the relative orientation of the enzyme in the composite structure obtained by ZDOCK (S4 Fig and S1 Table). Generally speaking, the former has a tightly bound conformation, a larger protein-protein contact interface and more interactions, including van der Waals forces, hydrogen bonds, electrostatic interactions and π-π stacking; random docking provides a smaller interface And less interaction.

An inspection of the 3D structure conformation revealed that the relative orientation of SpyTag/SpyCathcer and the enzyme in SpyRing complex was significantly different from that ZDOCK complexes (S4 Fig and S1 Table). Generally speaking, the SpyRing complex had a tightly bound conformation, a larger protein-protein contact interface and more interactions, including van der Waals forces, hydrogen bonds, electrostatic interactions and π-π stacking, while random docking gave smaller interface and less interaction.

## Discussions

It is well known that the interaction at the interface takes precedence over Cys-Cys, opposite charge (electrostatic interaction) and hydrophobicity, while the interface residues are mainly characterized by hydrophobicity, aromaticity and long side chains [47, 48]. Therefore, here we changed the electrostatic interaction to study the effect of changing the protein-protein interface interaction on the thermal stability of the enzyme. Electrostatic force may play an electrostatic guiding role in the process of protein recognition [49], which can promote the formation of protein-protein interactions and provide directionality [50]. Moreover, it can diffuse in a large range and improve the binding rate [51]. One of the unique features of SpyRing cyclization revealed by the simulation calculations in this study was the existence of electrostatic interactions that play a guiding and directional role in the entire process of docking SpyCatcher/SpyTag with enzyme. These interactions did not have to last until the end of the docking process, nor were they the most important forces for maintaining the protein-protein interfaces. But they could bring oppositely charged amino acids and their neighboring amino acids close to each other, thereby facilitating other interactions. This observation was

**Table 1. The change in binding energy of each component with increasing temperature (from 333 K to 348 K).** The unit of energy is kJ/mol.

|  | Van der Waal energy | Electrostatic energy | Polar solvation energy | nonpolar solvation energy | Binding energy |
|---|---|---|---|---|---|
| Wild type (333 K) | -36.6 ± 2.0 | -462.0 ± 16.8 | 333.4 ± 12.4 | -9.0 ± 0.3 | -175.5 ± 7.7 |
| Wild type (348 K) | -86.1 ± 3.1 | -164.6 ± 20.1 | 203.1 ± 15.7 | -15.0 ± 0.6 | -61.6 ± 9.2 |
| K122G (333 K) | -44.2 ± 1.7 | -941.7 ± 12.9 | 433.4 ± 9.2 | -11.7 ± 0.1 | -564.45 ± 5.6 |
| K122G (348 K) | -8.4 ± 0.8 | -290.7 ± 11.7 | 100.5 ± 9.0 | -3.0 ± 0.2 | -202.1 ± 3.4 |

consistent with previous studies [31]. It can be seen from the calculation results of binding energies that electrostatic energy contributes most to the total combined energy. This proved that the contribution of electrostatic interaction between two molecules at a distance of 10 Å was higher than other energy components. Therefore, for protein interface, mutations that cause changes in electrostatic interactions may have a greater impact.

Increasing temperature may also have a significant impact on the protein-protein interface interaction and binding energy. Here, we heated the SpyRing system by 15K, that is, from 333 K to 348 K, to study the interaction of SpyTag/SpyCather with the enzyme at higher temperature, and to determined the protein-protein interface when the system reached equilibrium after heating. S2 Table recorded and compared the interface interactions at the two temperatures, showing that the protein-protein interface decreased after heating (from 333 K to 348 K). Here the most important feature was the reduction of electrostatic interactions, while other interactions such as hydrogen bonds and van der Waals forces did not change significant; this might be because the increase in temperature increases the randomness of the above-mentioned electrostatic interactions, especially in the intermediate process of interface formation, which caused the ionic bond to break. In addition, from the perspective of interface energy, van der Waals energy, electrostatic energy, and nonpolar solvation energy decreased with increasing temperature, while the polar solvation energy increased; these would also lead to a decrease in total energy (Table 1).

In this study, mutations were introduced at the protein interface to change their interactions, so as to explore their influence on enzyme thermostability. We found that the key amino acids on the interface of the mutant protein change the interface interaction and lead to different results. In fact, from the perspective of enzyme engineering, in addition to adjusting electrostatic interactions, we may also consider changes in hydrophobic interactions. In addition, hot spot residues may also cause large change in the interface [52, 53]. In short, we can adjust the protein interface according to different needs, and finally achieve specific goals.

## Conclusions

In summary, molecular dynamics simulations were used to study protein-protein interactions in the SpyRing model system, in which the single-domain enzyme lichenase from *B. Subtilis* 168 was cyclized by SpyTag/SpyCather through a covalent reaction. Our calculations showed that SpyTag/SpyCather can form stable interface with the target enzyme, in which electrostatic interaction play an importance role. We further studied the mutations and their effects on enzyme stability by changing the interface electrostatic interactions, including the introduction of oppositely charged amino acids or neutral amino acids. Our simulations suggested the introduction of mutations at the protein-protein interface did have a significant impact on the interface interaction and binding energy, thereby affecting the thermal stability of the enzyme. Unlike random docking interaction, appropriate mutations in SpyRing will generate new interface interactions, which will increase the binding energy of the enzyme with SpyTag/SpyCather, thus improving the thermal stability of the enzyme. Our results highlighted the

importance of the protein-protein interface interaction between SpyTag/SpyCather and the target enzymes, which suggests that it might be useful to explore possible PPIs and their impact on the target enzyme through simulation when designing the SpyRing cyclization reaction.

## Supporting information

**S1 Fig. The RMSD and RMSF (the inset of S1 Fig) of lichenase with the T136K mutation.** The red arrows in the RMSF diagram refer to the active sites.
(TIF)

**S2 Fig. The RMSD and RMSF (the inset of S2 Fig) of lichenase with the K122E mutation.** The red arrows in the RMSF diagram refer to the active sites.
(TIF)

**S3 Fig. The RMSD and RMSF (the inset of S3 Fig) of lichenase with the D124G mutation.** The red arrows in the RMSF diagram refer to the active sites.
(TIF)

**S4 Fig.** The protein structure calculated by the ZDOCK program (left), in which green is lichenase, yellow is SpyTag, red is SpyCatcher, and the protein-protein contact interface shown in the box (right).
(PNG)

**S1 Table. The difference between the Tag/Catcher-enzyme interface of the simulated SpyRing complex structure and that of the random docking complex.** The amino acid sequence of the random complex was aligned with that of the SpyRing complex, so the same amino acid has the same sequence number, that is, the sequence number of the SpyRing complex.
(DOCX)

**S2 Table. The interface changes with increasing temperature (from 333 K to 348 K).**
(DOCX)

## Acknowledgments

We are grateful to the High Performance Computing Center of Nanjing Tech University for supporting the computational resources.

## Author Contributions

**Conceptualization:** Dangling Ming.

**Data curation:** Qi Gao.

**Funding acquisition:** Dangling Ming.

**Validation:** Qi Gao.

**Visualization:** Qi Gao.

**Writing – original draft:** Qi Gao, Dangling Ming.

**Writing – review & editing:** Dangling Ming.

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
