## [Decision Letter · Decision Letter 0]

19 Oct 2021

PONE-D-21-25882Protein-protein interactions enhance the thermal resilience of SpyRing-cyclized enzymes: a molecular dynamic simulation studyPLOS ONE

Dear Dr. Ming,

Thank you for submitting your manuscript to PLOS ONE. After careful consideration, we feel that it has merit but does not fully meet PLOS ONE’s publication criteria as it currently stands. Therefore, we invite you to submit a revised version of the manuscript that addresses the points raised during the review process.

We look forward to receiving your revised manuscript.

Kind regards,

Jie Zheng, Ph.D

Academic Editor

PLOS ONE

Journal Requirements:

This work was supported, in part, by the National Key Research and Development Program of China (Grant No. 2019YFA0905700, 2017YFC1600900) to DM. We are grateful to the High Performance Computing Center of Nanjing Tech University for supporting the computational resources.

This work was supported, in part, by the National Key Research and Development Program of China (Grant No. 2019YFA0905700, 2017YFC1600900). We are grateful to the High Performance Computing Center of Nanjing Tech University for supporting the computational resources.

This work was supported, in part, by the National Key Research and Development Program of China (Grant No. 2019YFA0905700, 2017YFC1600900). We are grateful to the High Performance Computing Center of Nanjing Tech University for supporting the computational resources.

5. Please amend your list of authors on the manuscript to ensure that each author is linked to an affiliation. Authors’ affiliations should reflect the institution where the work was done (if authors moved subsequently, you can also list the new affiliation stating “current affiliation:….” as necessary).

Reviewers' comments:

Reviewer's Responses to Questions

**Comments to the Author**

1. Is the manuscript technically sound, and do the data support the conclusions?

Reviewer #1: No

Reviewer #2: Partly

2. Has the statistical analysis been performed appropriately and rigorously? 

Reviewer #1: No

Reviewer #2: Yes

3. Have the authors made all data underlying the findings in their manuscript fully available?

Reviewer #1: No

Reviewer #2: Yes

4. Is the manuscript presented in an intelligible fashion and written in standard English?

Reviewer #1: No

Reviewer #2: No

5. Review Comments to the Author

Reviewer #1: This article applied MD simulations to study the protein-protein interactions and is aimed to demonstrate these interactions can enhance the thermal resilience of SpyRing-cyclized enzymes. They mutated some residues to study their role in the protein-protein interactions. However, I wouldn’t like to this paper to be published due to the following reasons:

1. Abstract is too long. The authors should remove some redundant research background and summarize the most important information of this work in abstract.

2. The arrangement and English writing of this manuscript should be greatly improved for better understand of the work.

3. Some typos in the method part should be corrected. For example, “Long range electrostatics with the cutoff distances of 1.0 nm and van der Waals were calculated using the particle mesh Ewald method (PME)” is not correct.

4. The author should mark the protein name in each snapshot figure.

5. From the configuration of protein-protein interactions, the author should calculate the distance change between protein and protein and between residue and residue as the function of simulation time to statistically estimate the dynamics change of the proteins and residues.

6. The author should do the structure alignment to compare the similar and different interactions between wild type and the mutants.

7. The author should analyze hydrogen bonding (including paired residues and occupancy of the hydrogen bonding) and interaction energies (values) between paired residues of the two protein to find key residues for the protein-protein interactions.

Reviewer #2: The SpyTag/SpyCatcher system is a useful technique to generate thermo-resistant enzymes, in which the N- and C- terminus of target enzymes were fused to the SpyTag and SpyCatcher peptides respectively. Here, Gao et al used MD simulation to study the dynamics of cyclized lichenase mediated by SpyTag/SpyCatcher and found a unique interface between lichenase and SpyTag/SpyCatcher. This result provides important insights on how the SpyTag/SpyCatcher system increases thermal stability of its target enzyme. I found this study is very interesting. However, I have two major concerns.

1. The authors state that this study provides new insights into the rational design of the SpyTag/SpyCatcher system. However, the target enzyme varies greatly in their structures and sequences. I don’t think the interface between SpyTag/SpyCatcher and lichenase here will be conserved in any other systems. Could the authors demonstrate why this feature will be a conserved property for SpyTag/SpyCatcher? Using Firefly luciferase as an example, will the SpyTag/SpyCatcher residues interacting with luciferase be the same as those interacting with lichenase?

2. Manuscript should be written in standard English before it could be published. However, I found grammatical errors throughout the manuscript. This manuscript should be re-written in standard English before it can be published in PLOS ONE. Two of obvious errors, but not all, are listed below.

On page 3, The basic mechanism is that when the Tag/Catcher genes (should be peptides, genes cannot be fused to protein) are fused to the N- and C- termini (terminus) of the enzyme, they spontaneously and rapidly react to form an irreversible iso-peptide bond between ASP (Asp117) of SpyTag and LYS (Lys31) of SpyCatcher, leading to the covalent cyclization of the enzyme.

On page 4, The cyclized enzyme has the optimum (optimal) temperature of 35℃, which was (a) 10℃ (increase) compared with the linear FLuc.

6. PLOS authors have the option to publish the peer review history of their article (what does this mean?). If published, this will include your full peer review and any attached files.

Reviewer #1: No

Reviewer #2: No

---

## [Author Response · Author response to Decision Letter 0]

30 Nov 2021

A separate file named "Replies to review comments_lichenase.doc" had been uploaded as the response to the reviewer and editor comments. Our responses are marked in red.

---

## [Editor Report · Decision Letter 1]

27 Jan 2022

Protein-protein interactions enhance the thermal resilience of SpyRing-cyclized enzymes: a molecular dynamic simulation study

PONE-D-21-25882R1

Dear Dr. Ming,

We’re pleased to inform you that your manuscript has been judged scientifically suitable for publication and will be formally accepted for publication once it meets all outstanding technical requirements.

Kind regards,

Jie Zheng, Ph.D

Academic Editor

PLOS ONE
---

## [Editor Report · Acceptance letter]

8 Feb 2022

PONE-D-21-25882R1 

Protein-protein interactions enhance the thermal resilience of SpyRing-cyclized enzymes: a molecular dynamic simulation study 

Dear Dr. Ming:

I'm pleased to inform you that your manuscript has been deemed suitable for publication in PLOS ONE. Congratulations! Your manuscript is now with our production department. 

Kind regards, 

on behalf of

Dr. Jie Zheng 

Academic Editor

PLOS ONE